DESY-XX-YYY

# QCD Axion Strings or Seeds?

**Simone Blasi,**[a,b] **Alberto Mariotti**[b]

[a] *Deutsches Elektronen-Synchrotron DESY, Notkestr. 85, 22607 Hamburg, Germany*

[b] *Theoretische Natuurkunde and IIHE/ELEM, Vrije Universiteit Brussel, & The International Solvay Institutes, Pleinlaan 2, B-1050 Brussels, Belgium*

*E-mail:* simone.blasi@desy.de, alberto.mariotti@vub.be

We study the impact of QCD axion strings on the cosmological history of electroweak (EW) symmetry breaking, focussing on the minimal KSVZ axion model. We consider the case of the pure SM Higgs potential as well as a simple scenario with a first order EW phase transition. We capture the effect of the Peccei–Quinn (PQ) sector within an effective–theory approach for the Higgs field, where the axion string core and the heavy PQ states are integrated out. The relevant parameters in this effective theory are controlled by the size of the portal coupling between the Higgs and the PQ scalar, and the mass of the PQ radial excitation. We determine the range of portal couplings for which the axion strings can strongly affect the dynamics of EW symmetry breaking. In the case of a first order EW phase transition, the strings can act as seeds by either catalyzing the nucleation of (non-spherical) bubbles, or leading to the completion of the phase transition by triggering a classical instability.

## 1 Introduction

The strong CP problem and the nature of the electroweak (EW) phase transition remain two important open questions in particle physics and cosmology. The QCD axion, arising from a spontaneously broken Peccei–Quinn (PQ) symmetry, represents one of the most compelling solution to the strong CP problem [1–6]. On the other hand, the EW phase transition and its modifications induced by physics beyond the SM (BSM) may be key for explaining the matter–anti matter asymmetry of our Universe [7–9], as well as generating an observable background of gravitational waves due to the nucleation of bubbles [10–13].

In the so called post–inflationary scenario where the PQ symmetry is spontaneously broken after inflation, the axion solution to the strong CP problem inevitably implies the formation of axion strings in the early universe according to the Kibble mechanism [14]. The string network survives until the time of the QCD crossover when the strings become the boundaries of axionic domain walls and the network starts to annihilate, provided that the axion model has a trivial domain wall number, $N_{\mathrm{DW}} = 1$ [15, 16].

Axion strings will then be present at the (earlier) time of EW symmetry breaking. It is therefore important to study how their presence can modify the nature of the EW

phase transition, which we investigate in this paper for the first time. This analysis is relevant both for the case in which the EW sector is the minimal one of the SM (so that EW symmetry breaking is a smooth crossover), as well as for BSM models with a first order EW phase transition. Examples of the latter include models with an extended Higgs sector including additional scalar singlets or doublets, see e.g. [17, 18].

The QCD axion strings we are focussing on are actually one example among the possible objects that can play the role of impurities for cosmological phase transitions. The relevance of inhomogeneous, or seeded, transitions in this context has been in fact recognized long time ago [19–26]. Topological defects, such as cosmic strings [24, 25, 27–32], domain walls [33–35], and monopoles [19, 20, 36–38] are among the most natural impurities arising from field theory. Compact objects, including (primordial) black holes, have been shown to affect the bubble nucleation rate as well [39–55]. Other possible sources are primordial density fluctuations [56] and energetic particle rays [57–60].

The impact of the axion strings on the EW phase transition is controlled by the size of the interactions between the PQ and the EW sector. In this paper we will consider the minimal KSVZ [61, 62] axion model where the SM Higgs is neutral under the PQ symmetry. Even in this case, there is no symmetry argument that can forbid a quartic interaction between the PQ complex scalar and the SM Higgs doublet. Such (neutral) portal is generic in the UV, and it implies an effective coupling between the Higgs field and the axion strings. Our goal will then be to systematically study the effect of the strings on the EW phase transition as a function of this portal interaction [1].

For the case of the pure SM + PQ theory we will determine under which conditions the axion strings can develop a sizable Higgs core where the EW symmetry is broken at temperatures well above the EW scale [2]. The mechanism is analogous to the one leading to superconducting strings in the abelian Higgs model [65–69]. Such EW core can modify the dynamics of the QCD axion strings in the early universe, as studied in recent numerical simulations [70].

For models in which the EW phase transition is first order, we will instead show how the axion strings can act as seeds catalyzing bubble nucleation. The relevant tunneling process interpolates between an axion string (typically with a vanishing Higgs core) and a new axion string with a Higgs condensate. This happens through the nucleation of $O(2)$ symmetric bubbles along the strings, as shown in Fig. 1 for one of our benchmark points, potentially with interesting new signatures (see e.g. [33–35] for the case of a domain–wall driven phase transition, and [71] for a recent study of bubbles with a reduced symmetry).

The other possible way in which strings can induce EW symmetry breaking in a model with a first order transition is, as first discussed in [24], via a classical instability of the string which is triggered at temperatures around the EW scale. We will describe in detail how this occurs for the QCD axion strings.

---

[1] In models where the Higgs is charged under the PQ symmetry, such as in DFSZ [3–6] type of models, one expects similar or even stronger effects.

[2] Axion string configurations with non–trivial Higgs and gauge boson profiles have been studied recently in [63, 64] for DFSZ models.

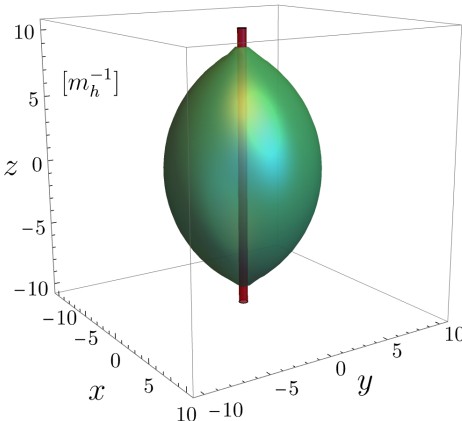

**Figure 1**: Three–dimensional representation of a critical bubble of broken electroweak symmetry seeded by the QCD axion string. The string is shown in red, and it is taken to be straight and aligned with the vertical $z$ direction. The Higgs bubble in green is nucleated around the string with a non–spherical shape, corresponding to the surface where the Higgs field is $h(r, z) \sim 25\,\mathrm{GeV}$ for illustration purposes. Detailed information is given in Sec. 5.3.

Let us also mention that, as one expects a large hierarchy between the EW scale and the PQ scale, our analysis will be based on an effective field theory (EFT) for the Higgs field where the heavy degrees of freedom (including the basic axion string) are integrated out [3]. Our EFT matches the known results for the SM + axion (or ALP) EFT, see e.g. [74–76], but additionally allows to take into account the presence of the axion string in a simple way. We will also comment on how the relevance of the different higher–dimensional operators in the ALP EFT is modified in the string background. We believe that our approach provides an efficient framework to study the dynamics of EW–scale states coupled to strings of large tension, which can be applied to many extensions of the SM.

This paper is organized as follows. In Sec. 2 we introduce our Lagrangian and comment on the different realizations depending on whether the EW phase transition is first order or not. We also present a brief overview of the possible QCD axion string solutions allowed by the model. In Sec. 3 we derive the EFT for the Higgs field in the string background, and carry out the relevant computations that are needed to study the thermal history of the Higgs sector. This is discussed in detail in Sec. 4 for the minimal SM + PQ scenario, and in Sec. 5 for a model with a first order EW phase transition. We conclude in Sec. 6.

## 2 Setup

Our setup consists of a complex scalar field $\Phi$ charged under a global $U(1)$ Peccei–Quinn symmetry coupled to the scalar sector of the Standard Model via a portal interaction of

---

[3]See [72, 73] for a similar approach in the context of branes and strings with fluxes.

coupling strength $\kappa$. This portal may be thought of as being effectively generated from loops of the KSVZ fermions, or it could be present in the theory already at the tree level.

The Lagrangian reads

$$\mathcal{L} = (\partial_\mu \Phi)^* \partial^\mu \Phi + (D_\mu \mathcal{H})^\dagger D^\mu \mathcal{H} - V_{\text{PQ}}(|\Phi|) - V_{\text{EW}}(|\mathcal{H}|; T) - 2\kappa \left( |\Phi|^2 - \frac{f_a^2}{2} \right) \left( |\mathcal{H}|^2 - \frac{v^2}{2} \right),$$

$$(2.1)$$

and we only consider scenarios with $\kappa > 0$. Here $V_{\text{PQ}}$ is the potential responsible for the PQ symmetry breaking,

$$V_{\text{PQ}} = -m^2 |\Phi|^2 + \eta |\Phi|^4 = \eta \left( |\Phi|^2 - \frac{f_a^2}{2} \right)^2,$$

$$(2.2)$$

where

$$\Phi = \frac{1}{\sqrt{2}} \rho(x) e^{i\alpha(x)}.$$

$$(2.3)$$

In (2.1), $V_{\text{EW}}(|\mathcal{H}|; T)$ is the potential energy of the Higgs sector, with the Higgs doublet such that $\langle \mathcal{H} \rangle = (0, h/\sqrt{2})$ in the vacuum at zero temperature, and we have included temperature corrections only in the purely EW part of the potential as we will be only considering temperatures $T < f_a$. For the moment we leave $V_{\text{EW}}$ unspecified, as we will be interested in two different scenarios depending on the electroweak phase transition (EWPT) being first or second order. The structure of the portal interaction is chosen such that at $T = 0$ the true vacuum of the theory is where $h = v$ and the axion decay constant is $f_a$.

We assume a post–inflationary PQ breaking scenario implying the formation of axion strings at high temperatures. Our focus will be the impact of the QCD axion strings on the cosmological history of the EW sector, depending on the size of the portal interaction $\kappa$. In this regard, we anticipate that the relevant quantities for our analysis will actually be the dimensionless ratio $\kappa/\eta$ and the mass of the radial mode of the PQ field, $m_\rho$.

We finally emphasize that the PQ–Higgs portal interaction is not protected by any symmetry and is in fact unavoidably generated by loops of the KSVZ fermions responsible for the mixed PQ–QCD anomaly. For instance, KSVZ fermions coupled to the PQ field with a yukawa interaction $y = M_\psi/f_a$ and to the SM only via QCD contribute to the portal by a three-loop diagram involving top quarks, which can be estimated as $\kappa_{\text{rad}} \sim 10^{-5}(M_\psi/f_a)^2$. Larger contributions are expected if the KSVZ fermions are charged under the EW group. In addition, depending on the specific UV theory one generically expects other contributions to this coupling from physics at scales higher (or equal) than $f_a$. For this reason, will treat the portal coupling $\kappa$ (as well as the PQ self–quartic $\eta$) as a free parameter in our analysis.

## 2.1 Scalar potential and its extrema

As mentioned above, we will consider two possible scenarios for the EW sector, which we implement by adopting different shapes for the EW potential $V_{\text{EW}}$ in (2.1):

- In the first case we stick to the SM potential, including leading thermal corrections in the high–$T$ expansion:

$$V_{\text{EW}} = V_{\text{SM}} \equiv -\frac{1}{2} \left( \mu^2 - c_h T^2 \right) h^2 + \frac{1}{4} \lambda h^4$$

$$(2.4)$$

where $c_h \simeq 0.4$ in the SM. Here we define $T_{\text{EW}}$ as the temperature at which the Higgs mass is vanishing.

- In the second case, we consider a potential that serves as a benchmark for scenarios with a first order EW phase transition [77] with a barrier between the EW preserving minimum ($h = 0$) and the EW breaking vacuum at all temperatures:

$$V_{\text{EW}} = V_\delta \equiv -\frac{1}{2}\left(\mu^2 - c_h T^2\right)h^2 + \frac{\delta}{3}\frac{m_h^2}{v^2}h^3 + \frac{1}{4}\lambda h^4. \qquad (2.5)$$

Here we take the same value of $c_h$ as in the SM for definiteness, and $\delta < 0$ determines the height of the barrier. This deformation reduces to the SM potential for $\delta = 0$. For a given value of $\delta \neq 0$, the other parameters are chosen to give the correct Higgs mass and vacuum expectation value (VEV),

$$\mu^2 = \frac{m_h^2}{2}(1 + \delta) \qquad \lambda = \frac{m_h^2}{2v^2}(1 - \delta), \qquad (2.6)$$

with $m_h = 125\,\text{GeV}$ and $v = 246\,\text{GeV}$. We finally define $T_c$ as the critical temperature where the two minima are degenerate.

Our model in (2.1) is then such that at $T = 0$ there is a global minimum (A) where the EW and PQ symmetries are spontaneously broken. For finite temperatures below the critical temperature $T_c$, or $T_{\text{EW}}$ in the SM case, this point remains the global minimum of the scalar potential, and it has the form

$$\text{A}: \quad \left\{\rho = \sqrt{f_a^2 + \frac{\kappa}{\eta}\left(v^2 - v^2(T)\right)},\ h = v(T)\right\} \qquad (2.7)$$

where $v(T = 0) = v$.

At high temperatures the point A becomes either a local minimum or a saddle, while the global minimum is

$$\text{B}: \quad \left\{\rho = \sqrt{f_a^2 + \frac{\kappa}{\eta}v^2} \equiv \tilde{f}_a,\ h = 0\right\}, \qquad (2.8)$$

where the EW symmetry is unbroken, and the axion decay constant is slightly shifted by corrections of the size of the EW scale to $\tilde{f}_a \simeq f_a$. There are two other extrema in this potential: the origin of field space, which is a maximum, and a saddle where ($\rho = 0, h \sim \kappa/\lambda\, f_a$). This structure of the scalar manifold is preserved if the portal coupling is small enough, $\kappa/\eta \lesssim 1$.

Let us finally notice that the structure of the Lagrangian (2.1), and in particular the portal interaction between the Higgs and the PQ field, implies a certain degree of fine tuning of the electroweak scale. In fact, one would expect corrections to the Higgs mass of order $\delta m_h^2 \sim \kappa f_a^2$ which are better rewritten as $\delta m_h^2 \sim (\kappa/\eta)\, m_\rho^2$, where the mass of the radial PQ excitation is given by $m_\rho = \sqrt{2\eta}\tilde{f}_a$ in the B vacuum. As we shall see, the axion strings will be affecting the EW phase transition for $\kappa/\eta \gtrsim 10^{-3}$, so that scenarios with $m_\rho$ heavier than a few TeV require accidental cancellations, or some additional stabilization

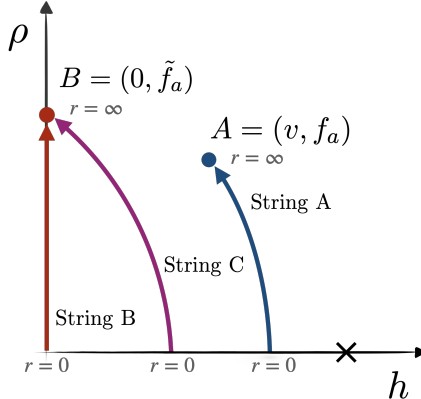

**Figure 2**: Cartoon of the path in field space of the various string solutions in the EW plus PQ model. In addition to the standard string $B$ where the Higgs is identically vanishing, there can be two type of strings with a Higgs core, ending asymptotically in the EW preserving vacuum (string C) or in the EW breaking vacuum (string A).

mechanism. In the following, we will remain agnostic about the (possible) fine tuning of the electroweak scale, and consider $\kappa/\eta$ and $m_\rho$ as free parameters.

In the rest of this section we provide an overview of the QCD axion string solutions that are allowed by the Lagrangian in (2.1). More details as well as a quantitative analysis of these solutions will be given in Sec. 3.

## 2.2 Overview of string solutions

Static string configurations are obtained as classical solutions to the equations of motion:

$$
\begin{aligned}
&\rho''(r) + \frac{\rho'(r)}{r} - \frac{\rho(r)}{r^2} + m^2\rho(r) - \eta\rho^3(r) - \kappa\left[h(r)^2 - v^2\right]\rho(r) = 0, \\
&h''(r) + \frac{h'(r)}{r} - V'_{\text{EW}}(h;T) - \kappa\left[\rho^2(r) - f_a^2\right]h(r) = 0,
\end{aligned}
\tag{2.9}
$$

where we have assumed cylindrical symmetry for $\rho$ and $h$, and the string is taken to be straight along the $z$ axis. Here $r$ is the radial coordinate on the plane orthogonal to the string, and the axion field, namely the phase $\alpha$ in (2.3), winds a single time,

$$
\alpha = n\,\theta,
\tag{2.10}
$$

where $\theta$ is the angle in physical space around the string, and $n = 1$. The axion winding provides the centrifugal force in the first line of (2.9). The equations of motion for the phase $\alpha$ are trivially satisfied when $\rho = \rho(r)$ and $\alpha = \theta$. Notice that no gauge background is turned on by the space–dependent Higgs profile $h(r)$, as this can be taken to be aligned along a constant direction in SU(2)×U(1) consistently with our ansatz $\mathcal{H}(r) = (0, h(r)/\sqrt{2})$. This differs for instance from the EW string solutions found in Ref. [63] (see also [78]) where the Higgs doublets have a non–trivial winding.

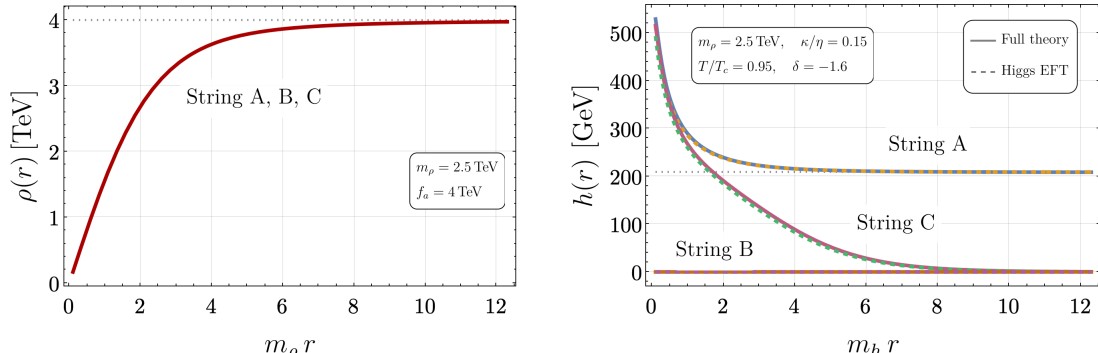

**Figure 3**: **Left:** Profile of the PQ field along the radial direction perpendicular to the string for the string B, as well as for string C and A as they differ by negligible EW scale corrections. **Right:** Profile of the Higgs field for string A, B and C. The profiles of string A and C have structure at all scales $r \lesssim 1/m_h$ and are characterized by a different asymptotic value of the Higgs at large $r \gg 1/m_h$.

All possible string configurations restore the PQ symmetry at the center of the string,

$$\rho(0) = 0. \tag{2.11}$$

Depending on the other boundary conditions, three types of strings are possible, that we label string A, B, and C. A cartoon of these configurations is shown in Fig. 2 as a path in field space $(h(r), \rho(r))$ starting from the core at $r = 0$ and ending in one of the minima at $r = \infty$. We anticipate that the $\rho$ profiles for string A, B, C differ only by small corrections of the order of the EW scale. This is because of the assumed hierarchy $m_h/m_\rho \ll 1$ as well as the moderate portal couplings, $\kappa/\eta \lesssim 1$. The different string solutions are then characterized by the Higgs profile, which can be trivial as in string B ($h \equiv 0$), or develop a non–vanishing core as in string A and C, as we now discuss.

**String B** This is the simplest (and most standard) solution to our system (2.9) where the fields approach the vacuum B arbitrarily far from the string. The corresponding boundary conditions are

$$\rho(\infty) = \tilde{f}_a. \qquad h(\infty) = 0. \tag{2.12}$$

These conditions do not uniquely identify string B, which is defined by further requiring that the Higgs field is identically vanishing,

$$h(r) \equiv 0. \tag{2.13}$$

The equations of motion (2.9) then simplify to

$$\rho''(r) + \frac{\rho'(r)}{r} - \frac{\rho(r)}{r^2} + \tilde{m}^2 \rho(r) - \eta \rho^3(r) = 0, \quad \tilde{m}^2 = m^2 + \kappa v^2, \tag{2.14}$$

and a typical profile can be seen in the left panel of Fig. 3.

The tension of string B, namely its mass per unit length, can be calculated as

$$\mu_B \simeq (\pi \log(R m_\rho) + c)\, \tilde{f}_a^{\,2}, \tag{2.15}$$

where $R > 1/m_\rho$ is an infrared cutoff matching the typical string curvature in a realistic network, and $c = \mathcal{O}(1)$ from the numerical integration. As we can see, the tension is logarithmically divergent. This is a special feature of global strings as opposed to local strings, in which the gauge fields ensure a faster convergence to zero of the scalar gradients.

String B corresponds to the standard QCD axion string, where the Higgs field is purely a spectator, and it is the only solution for vanishing portal coupling ($\kappa = 0$). It is also the one usually studied in numerical simulations of QCD axion string networks in the expanding universe [79–86].

For a non–vanishing interaction with the Higgs sector, however, string B is not guaranteed to be stable. In order to assess this, one has to look at the spectrum of perturbations around the string background that solves (2.14). At the linear order in the perturbations, the fluctuations in the PQ and the Higgs fields are decoupled. The equation for the fluctuation $\sigma(r)$ around the $\rho(r)$ background can be derived from (2.14) and reads

$$\left[ -\frac{d^2}{dr^2} - \frac{1}{r}\frac{d}{dr} + \frac{1}{r^2} - \tilde{m}^2 + 3\eta\rho^2(r) \right] \sigma(r) = \Omega^2 \sigma(r). \tag{2.16}$$

This equation allows also for a bound state solution with $\sigma(\infty) = 0$ and mass given by

$$\Omega^2 = 1.63.. \, \tilde{m}^2. \tag{2.17}$$

The bound state is localized around $r = 0$ on scales $\sim 1/\tilde{m}$, and $\Omega^2 > 0$ guarantees that the string is classically stable for perturbations around the PQ radial mode.

The overall stability of the string is then controlled by the mass of the Higgs fluctuations, which can become negative depending on the temperature and the portal coupling. While a detailed analysis is postponed to Sec. 3.3, here we simply notice that the existence of this instability suggests that the same boundary conditions (2.12) can support another type of string solution where the Higgs profile is non vanishing.

**String C** When $\kappa \neq 0$, it is indeed possible to find solutions to the full system of equations in (2.9) with the same boundary conditions as in (2.12), where the Higgs profile is non trivial. For the EW potential as in the SM, this is naturally realized when the string B becomes unstable for Higgs fluctuations above the critical temperature ($T > T_{\rm EW}$) when the point B in field space is still the global minimum. The string C can however be found also for $T < T_c$ if the EW potential has a local minimum at the origin, as in scenarios with a first order EW phase transition.

A typical Higgs profile for this string solution is shown in the right panel of Fig. 3. As we can see, it has structure at all scales between $1/m_\rho$ and $1/m_h$, starting from a potentially large Higgs core and vanishing at distances $r \gg 1/m_h$. EW symmetry is therefore broken only at distances $\sim 1/m_h$ around the string, and restored far from it. The profile of the PQ field is similar to the one of string B, as shown in the left panel of Fig. 3.

**String A** The last possible solution is characterized by $h(r)$ and $\rho(r)$ ending in the EW–breaking vacuum A far from the string core at $r = \infty$,

$$\rho(\infty) = f_a, \qquad h(\infty) = v(T). \tag{2.18}$$

String A is absolutely stable at low temperatures independently of the actual shape of the EW potential $V_{\text{EW}}$. This configuration should then be the end point of any phenomenologically viable cosmological history. When the EW potential is the SM one, the point A exists as a minimum only for $T < T_{\text{EW}}$, and so does string A.

A typical Higgs profile is shown in the right panel of Fig. 3. The field reaches a large value inside the string core at scales $r \sim 1/m_\rho$ similarly to string C. This increases for heavier $m_\rho$ and larger portal couplings, as we will further detail in Sec. 3. At distances $r \gg 1/m_h$ the Higgs profile approaches its value in A given by $h = v(T)$. The profile shows non–trivial structure at all scales between $1/m_\rho$ and the EW scale.

The $\rho(r)$ profile is shown in the left panel of Fig. 3. As mentioned, this profile is largely independent of the Higgs field and shows the same features in all string solutions: it varies on scales $\sim m_\rho$ and reaches the appropriate asymptotic value given here by (2.18).

## 3 Effective field theory for the EW sector

In this section we derive an effective field theory (EFT) approach to describe the physics below the scale of the radial PQ excitation $m_\rho$, assuming that $m_h/m_\rho \ll 1$ as well as $\kappa/\eta \lesssim 1$. In particular, we will focus on how the presence of the axion string can be included in this EFT with the aim of reproducing all the relevant physics in the EW sector regardless of the explicit profile of the PQ field.

Let us then consider a straight axion string along the $z$ axis with the profile of string B where the Higgs background is trivial, $h \equiv 0$. This intersects the orthogonal plane at the origin $r = 0$. Let us first focus on the region of space very far from the string core, namely $r \gg \epsilon \sim 1/m_\rho$. In this region the radial PQ field is approaching the B vacuum, $\rho = \tilde{f}_a$. We can then expand the action around this point,

$$\rho = \tilde{f}_a + \delta\rho, \tag{3.1}$$

and derive the equations of motion for $\delta\rho$ from (2.1):

$$\left[ \Box + m_\rho^2 - (\partial_\mu \alpha)^2 + 2\kappa|\mathcal{H}|^2 \right] \delta\rho = -\tilde{f}_a \left( 2\kappa|\mathcal{H}|^2 - (\partial_\mu \alpha)^2 \right). \tag{3.2}$$

In the string background under consideration $\alpha = \theta$ so that $(\partial_\mu \alpha)^2 = -1/r^2$, but we shall leave this implicit.

Terms on the LHS other than $m_\rho^2$ will introduce higher–order operators that are always suppressed by powers of $m_\rho$, and are neglected for simplicity in what follows. Substituting this back in the Lagrangian (2.1) we obtain the following leading–order action for the Higgs sector:

$$S_{r>\epsilon} = \int_{r>\epsilon} d^4x \left\{ \frac{f_a^2}{2}(\partial_\mu \alpha)^2 + (D_\mu \mathcal{H})^\dagger D^\mu \mathcal{H} - V_{\text{eff}}(|\mathcal{H}|;T) - \frac{\kappa}{\eta}(\partial_\mu \alpha)^2 |\mathcal{H}|^2 \right\}, \tag{3.3}$$

where

$$V_{\text{eff}}(|\mathcal{H}|;T) = V_{\text{EW}}(|\mathcal{H}|;T) - \frac{\kappa^2}{4\eta}(2|\mathcal{H}|^2 - v^2)^2. \tag{3.4}$$

The threshold correction to the Higgs potential is reabsorbed in a shift of the bare quartic coupling, so that the Higgs mass and the EW vev match the observed values. The only relevant modification to the EW Lagrangian up to $\mathcal{O}(1/m_\rho)$ corrections is then the axion–Higgs portal (see e.g. [87] for a study of this operator) whose size is controlled by the ratio $\kappa/\eta$. We have checked that other choices of the background around which $\rho$ fluctuations are integrated out lead to the same result, in particular the choice $\rho = \rho(r) + \delta\rho$, with $\rho(r)$ the profile of string B.

Let us now consider the region around the string core. Here the field $\rho$ is varying rapidly on scales $1/m_\rho$. For small values of the portal coupling, the hierarchy between $m_\rho$ and $m_h$ implies that the Higgs remains frozen at these scales. We may then describe the effect of the string as a localized Dirac $\delta$–potential. To this end we include an additional potential term in the effective theory of the form

$$S_{r=\epsilon} = \int d^4x\, T(|\mathcal{H}|)\delta^{(2)}(r - \epsilon), \tag{3.5}$$

where $T(|\mathcal{H}|)$ is a function of the Higgs field to be determined by the matching with the UV action. For our simple Higgs portal model one finds

$$T(|\mathcal{H}|) = -2\pi \int_0^\epsilon r\, dr\, \kappa(\rho^2(r) - f_a^2)|\mathcal{H}|^2 + \mathcal{O}(\epsilon), \tag{3.6}$$

where the only term in the potential that survives the limit of $1/\epsilon \sim m_\rho \to \infty$ is in fact the portal interaction. The integral is more clearly written as

$$T(|\mathcal{H}|) = 2\pi\frac{\kappa}{\eta}C(\epsilon)|\mathcal{H}|^2, \quad C(\epsilon) = \int_0^\epsilon r\,dr\, \eta(f_a^2 - \rho^2(r)). \tag{3.7}$$

The dimensionless function $C(\epsilon)$ can be evaluated numerically by substituting the actual string profile $\rho(r)$. One finds for instance $C \simeq 1.2$ for $\epsilon = 2\sqrt{2}/m_\rho$. As we can see, the precise shape of the string profile at scales $m_\rho$ does not matter, and the overall strength of the interaction with the string is encoded in the coefficient of the Dirac $\delta$–potential.

In summary, our effective action for the EW sector takes the form

$$S_{\text{EFT}} = \int_{r\geq\epsilon} d^4x \left\{ \frac{f_a^2}{2}(\partial_\mu\alpha)^2 + (D_\mu\mathcal{H})^\dagger D^\mu\mathcal{H} - V_{\text{EW}}(|\mathcal{H}|; T) + \right.$$
$$\left. - \frac{\kappa}{\eta}\left[(\partial_\mu\alpha)^2 - 2\pi\, C(\epsilon)\, \delta^{(2)}(r - \epsilon)\right]|\mathcal{H}|^2 \right\}. \tag{3.8}$$

This is consistent with the standard result obtained by integrating out the heavy PQ states, see e.g. [76], but it differs due to the $\delta$–potential coming from the string, and the fact that $\alpha$ has a non–trivial space dependence according to the winding. It is interesting to notice that even though the axion–Higgs portal is technically a dimension–six operator, the $1/f_a^2$ suppression is lifted by the non trivial axion configuration around the string, $a = \theta f_a$. For this reason, this operator is much more important than the other pure SMEFT (namely, axion independent) dimension–six operators that are generated at the same order.

When the KSVZ fermions are taken into account, additional operators need to be included in the low energy theory, notably the axion coupling to QCD, as well as possible couplings to EW gauge bosons and SM fermions. These interactions do not have a direct impact on the EW phase transition and are neglected in what follows.

The presence of the $\delta$–potential in (3.8) imposes a matching condition for the Higgs field at $r = \epsilon \sim 1/m_\rho$. By taking $\mathcal{H}(r) = (0, h(r)/\sqrt{2})$, the equation of motion implies

$$\epsilon\, h'(\epsilon) = -C(\epsilon)\frac{\kappa}{\eta}h(\epsilon). \tag{3.9}$$

This matching condition could be derived directly from the equations of motion of the UV theory by performing the $\int_0^\epsilon r dr$ integration of the Higgs equation.

We finally stress that, as anticipated, only the dimensionless ratio $\kappa/\eta$ and the UV scale $\epsilon \sim 1/m_\rho$ enter the effective action (3.8). Therefore, these two parameters are enough to capture all the implications of the QCD axion strings for the EW sector.

## 3.1 String solutions in the EFT

In this section we discuss how the Higgs profiles introduced in Sec. 2.2 for string A, B and C can be obtained within the effective theory (3.8). We will additionally comment on the stability of these solutions.

**String B and its stability**  Let us consider the profile of string B, which is trivial as far as the Higgs field is concerned, $h \equiv 0$. This however does not ensure that this solution is stable once the Higgs fluctuations around this background are taken into account [4]. To do so, we consider the EFT (3.8) and study the eigenvalue problem for the simplest configuration with $h(r, \theta) = h(r)$:

$$\left[-\frac{d^2}{dr^2} - \frac{1}{r}\frac{d}{dr} - (\kappa/\eta)\frac{1}{r^2} + V''_{\mathrm{EW}}(0; T)\right] h(r) = \omega^2 h(r), \tag{3.10}$$

where we have used that $(\partial_\mu \alpha)^2 = -1/r^2$ in the string background, and $\mathcal{H}(r) = (0, h(r)/\sqrt{2})$. If the lightest excitation has a mass $\omega^2 > 0$ then the string B is classically stable, while if $\omega^2 < 0$ the string will classically develop a Higgs core [5].

Eq. (3.10) allows for a bound state profile given in terms of a modified Bessel function,

$$h(r) \propto K_{i\sqrt{\kappa/\eta}}(\tilde{\omega}r), \qquad \tilde{\omega}^2 = V''_{\mathrm{EW}}(0; T) - \omega^2. \tag{3.11}$$

Crucially, the value of $\omega$ or equivalently $\tilde{\omega}$ is obtained by imposing the matching condition (3.9) on the profile (3.11). By expanding for $\kappa/\eta \lesssim 1$ and for $\tilde{\omega} \ll m_\rho$, one obtains:

$$\omega^2 = V''_{\mathrm{EW}}(0; T) - \frac{1}{2}m_\rho^2 f(\kappa/\eta), \tag{3.12}$$

---

[4]We have already shown in Sec. 2.2 that the fluctuations of the $\rho$ field are in fact stable.

[5]The final configuration following this instability depends on the temperature at which the instability occurs as well as on other details of the Higgs potential.

with

$$f(\kappa/\eta) = \exp\left\{-\frac{\pi}{\sqrt{\kappa/\eta}} - \gamma_{\rm E} + 2C(\epsilon)\right\}, \quad (\kappa/\eta \lesssim 1). \tag{3.13}$$

The quantity in (3.12) gives the mass of the Higgs bound state localized around the string, and it can be used to probe how strongly the string affects the EW sector. In particular, one can identify a decoupling limit in which the mass of this bound state is approximately the same as the Higgs mass far from the string given by $V_{\rm EW}''(0;T)$, so that the Higgs dynamics is essentially unaffected by the presence of the string. From the expression in (3.13), we see that the decoupling limit is approached fast for $\kappa/\eta \ll 1$ due to the exponential dependence on $(\kappa/\eta)^{-1/2}$. We notice that, for comparison, a $U(1)$ gauge string would decouple even faster, namely exponentially with $(\kappa/\eta)^{-1}$ [68, 69]. This difference can be shown to originate from the axion–Higgs portal term.

The value of $\tilde{\omega}$ determines the spread of the bound state on the plane orthogonal to the string as $\sim 1/\tilde{\omega}$. It is then clear that our approximation breaks down whenever $\tilde{\omega} \sim m_\rho$, as the bound state becomes sensitive to the actual profile of string B at short distances. From (3.12) we see that this occurs when $\kappa/\eta \sim 1$. Even though this case goes beyond the validity of our effective approach, the mass of the Higgs bound state can still be obtained analytically by performing a different (harmonic) approximation of the string B, yielding an expression equivalent to (3.12) but with the $f$ function replaced by $\tilde{f}$,

$$\tilde{f}(\kappa/\eta) = \frac{\kappa}{\eta} - 2b\sqrt{\frac{\kappa}{\eta}}, \quad b = 0.58\ldots \quad (\kappa/\eta \gtrsim 1). \tag{3.14}$$

This region is not very relevant for our work and will not be considered further.

With the knowledge of $\omega^2$ in (3.12) one can easily identify the regions in the $(\kappa/\eta)$–$m_\rho$ parameter space where the string B is (un)stable as a function of the temperature. In particular one can determine $T_r^B$ defined as the *rolling* temperature of the string B such that $\omega^2(T_r^B) = 0$, implying that for $T > T_r^B$ the string is stable, and unstable otherwise.

**Strings with non–trivial Higgs core** String A and C (when they exist) are characterized by a possibly very large Higgs core with $h(0) \gg v$ which decreases at large distances. String C may also be seen as a deformation of string B given that it asymptotes to the same vacuum far from the core.

Both the A and C profiles can be obtained from the Higgs equation of motion according to the EFT in (3.8):

$$h''(r) + \frac{h'(r)}{r} + (\kappa/\eta)\frac{h(r)}{r^2} = V_{\rm EW}'(h;T), \quad \epsilon\, h'(\epsilon) = -C(\epsilon)\frac{\kappa}{\eta}h(\epsilon), \tag{3.15}$$

where we have included the matching (3.9) at the string core, $\epsilon \sim 1/m_\rho$, and asymptotic boundary conditions given by

$$h(\infty) = v(T) \text{ [string A]}, \qquad h(\infty) = 0 \text{ [string C]}. \tag{3.16}$$

Solutions to (3.15) can be found numerically via shooting techniques. One can for instance scan different values of $h(\epsilon)$, which also fix the slope $h'(\epsilon)$ due to the matching,

until the Higgs profile approaches the required value $h(\infty)$. This method is particularly suited for string A. For string C we find more convenient to rely on an approximate solution to (3.15) at large distances, where $h \approx 0$ due to the boundary condition. The EW potential can then be linearized, and a solution is obtained as

$$h(r \gg 1/m_h) \simeq k \cdot K_{i\sqrt{\kappa/\eta}}(m_h r). \tag{3.17}$$

One can then scan different values of the constant $k$ at $r \gg 1/m_h$ until $h(\epsilon)$ satisfies the matching condition (3.9).

A comparison of the profiles obtained by solving the system of equations in the complete theory, (2.9), and the ones obtained within the EFT is shown in the left panel of Fig. 3. As we can see, the two methods quantitatively agree in determining the string profiles, thus providing a non–trivial cross check of our calculations.

## 3.2 String tension

We can also use the effective action (3.8) to calculate the difference in tension between our string solutions. This provides information on which string is actually the lightest and thus most stable solution. In addition, the difference in tension between the strings can be used to estimate the tunneling rate among them as we shall see in Sec. 5.3.

Notice that since (3.8) is obtained by integrating out fluctuations around the string B, the effective action will directly give the difference in tension between string B and the string solution with a non–trivial Higgs core $h(r)$.

Let us then focus on string A at zero temperature, where $h(\infty) = v$. The main contribution to the tension difference comes from the $\delta$–potential and from the axion–Higgs portal as the latter is logarithmically enhanced. One finds:

$$\Delta\mu \approx -\pi\frac{\kappa}{\eta}\int_\epsilon^R rdr\left\{\frac{1}{r^2} + 2\pi\,C(\epsilon)\delta^{(2)}(r-\epsilon)\right\}h(r)^2, \tag{3.18}$$

where $R$ is an infrared cutoff, and we have used the fact that the integral over $z$ in (3.8) simply gives the length of the string. Taking into account that $h(r \gg 1/m_h) \simeq v$, we obtain

$$\Delta\mu \approx -\pi\frac{\kappa}{\eta}\left\{C(\epsilon)\,h(0)^2 + \ln(R\,m_h)v^2 + \int_\epsilon^{m_h^{-1}} dr\frac{h(r)^2}{r}\right\}. \tag{3.19}$$

The first term comes from the $\delta$–potential, and we have used $h(\epsilon) \approx h(0)$. The second term takes into account the long–distance contribution from the axion–Higgs portal in the region where $h(r) \approx v$, and it is sensitive to the IR cut–off due to the global nature of the axion string. The last term is the most difficult to evaluate without the precise knowledge of the $h(r)$ profile, but it is expected to scale as $\sim \ln(m_\rho/m_h)\,h(0)^2$.

As we can see, $\Delta\mu$ is mostly controlled by the value of the Higgs core at center, $h(0)$. This can actually be obtained by solving (3.15) numerically with the appropriate boundary conditions. In Fig. 4 we show $h(0)/v$ for string A at $T = 0$ as a function of $\kappa/\eta$ for several values of $m_\rho$, assuming the EW potential to be the one in the SM (similar results are obtained when including the deformation in (2.5)). For very small portals, which we

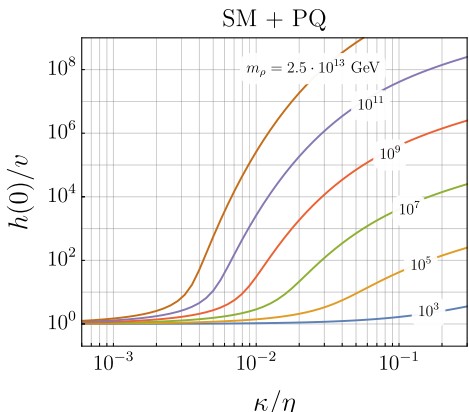

**Figure 4**: Higgs core in the axion string A for different values of $m_\rho$ as a function of $\kappa/\eta$. The EW potential is taken to be the one in the SM at $T = 0$. The size of the Higgs core provides an insight of the axion string decoupling with $\kappa/\eta \ll 1$, as well as the difference in tension between string A and B (see text).

quantify as $\kappa/\eta \lesssim 10^{-3}$, the Higgs condensate in the string approaches its value far from the string, namely $h(0) \approx v$, even for very large values of $m_\rho$. This is consistent with the exponential decoupling of the axion string as a function of $\sqrt{\kappa/\eta}$ found in (3.12). On the other hand, for larger values of $\kappa/\eta$ the condensate can be several orders of magnitude larger than the EW scale depending on $m_\rho$.

Let us conclude this section by noticing that for the moderate values of $\kappa/\eta \lesssim 0.1$ we are interested in, the difference in tension between string A and B is actually small when compared to the absolute tension of string B in (2.15), $\Delta\mu/\mu_B \ll 1$, as we always have $h(0) < f_a$. This holds also for the difference in tension between string B and C (when the latter exists).

### 3.3 The 1+1 theory on the axion string

It can be useful to make a further step and derive a lower dimensional EFT for the Higgs fluctuations that live on the string B. These are determined by the following ansatz

$$h(z, r) = h_0(t, z)h(r), \tag{3.20}$$

where $z$ is the coordinate on the string, and $h(r)$ is the profile in (3.11). Notice that from the very beginning we are limiting our analysis to a single bound state with no angular dependence. Therefore, this effective theory works only when a large hierarchy exists between $\omega^2$ and $m_h^2$. By integrating out the radial direction $r$ in (3.8), we obtain a 1+1 action for $h_0$:

$$S_{1+1}[h_0] = \int dz dt \left\{ \frac{1}{2}(\partial_\mu h_0)^2 - \tilde{V}(h_0) \right\}, \quad \tilde{V}(h_0) = \frac{1}{2}\omega^2 h_0^2 - \frac{1}{3!}c_3 h_0^3 + \frac{1}{4!}c_4 h_0^4. \tag{3.21}$$

The $c_{3,4}$ coefficients are calculated from the overlap integrals involving the profile of the Higgs bound state:

$$c_3 = -V_{\text{EW}}^{(3)}(0;T) \cdot \int_0^\infty 2\pi r \, dr \, h^3(r), \quad c_4 = V_{\text{EW}}^{(4)}(0;T) \cdot \int_0^\infty 2\pi r \, dr \, h^4(r), \qquad (3.22)$$

where $(n)$ indicates the $n$–th derivative. This lower dimensional theory will be used in Sec. 5.3 to evaluate the tunneling action in a certain range of couplings, analogous to the case of domain wall seeds [33].

## 4 Thermal history of SM plus PQ

We can apply the analysis in the previous section to study the effect of the axion strings on the cosmological history of the Higgs sector, focussing first on the most minimal scenario in which the Higgs potential is the one of the SM, $V_{\text{EW}} = V_{\text{SM}}$.

At temperatures $T \sim f_a$ when the $U(1)_{\text{PQ}}$ is spontaneously broken, axion strings are formed and will soon reach a scaling regime where the number of strings per Hubble volume is of order $\xi \sim O(1-100)$ [79–84].

At high temperatures the strings are the standard ones of type B, that is without a Higgs core, due to the large and positive thermal mass of the Higgs. While the temperature drops, the string B will eventually develop an instability along the Higgs direction and form a core where the EW symmetry is broken, thus transforming into string C. This occurs at the *rolling* temperature of string B defined by $\omega^2(T_r^B) = 0$. As the mass of the lightest Higgs fluctuation around the string is always smaller than the Higgs mass in the bulk, see (3.12), one always has $T_r^B > T_{\text{EW}}$, where $T_{\text{EW}} \simeq 140$ GeV marks approximately the SM cross-over temperature in the high–temperature approximation. Depending on the actual value of $\kappa/\eta$, $T_r^B$ can be approximately equal to $T_{\text{EW}}$ (for small portals close to the decoupling limit) or much larger. Contours of $T_r^B/T_{\text{EW}}$ are shown in the left panel of Fig. 5 as a function of $m_\rho$ and $\kappa/\eta$. The white region indicates where the axion strings are effectively decoupled from the Higgs sector, and the thermal history is basically the same as in the SM: in this case the Higgs field develops a vacuum expectation value inside the string and in the bulk approximately at the same temperature, $T_r^B \approx T_{\text{EW}}$.

In the blue region instead, the EW symmetry is broken inside the string at much higher temperatures than in the bulk, as indicated by the contours. In particular, there exists a potentially very large range of temperatures, $T_{\text{EW}} < T < T_r^B$, where the only stable string solution is of type C with a non–vanishing Higgs condensate. As soon as the temperature drops below $T_{\text{EW}}$, this configuration will smoothly deform into a string A following the EW crossover in the bulk. The size of the Higgs condensate inside the string has been shown in Fig. 4 as a function of $\kappa/\eta$. The calculation was performed at $T = 0$ for string A, but it nevertheless gives the right order of magnitude for string C as well at all temperatures below the instability, $T \lesssim T_r^B$.

Let us conclude this section by noticing that the spontaneous breakdown of the EW symmetry at the core of string C at temperatures much above the EW scale can potentially modify the QCD axion string dynamics in the early universe, e.g. in terms of friction with

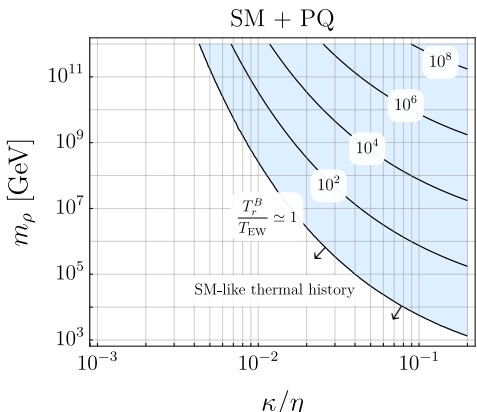

**Figure 5**: Contours of $T_{\text{roll}}^B$ as a function of the PQ radial mode mass $m_\rho$ and the portal coupling $\kappa/\eta$. For $T_{\text{roll}}^B \gg T_{\text{EW}}$ the cosmological history is modified with axion strings developing an EW symmetry breaking core at high temperatures (light blue region). In the white region, the thermal history is practically indistinguishable from the pure SM.

the thermal plasma, or other properties of the network such as particle production. A study of these features is left for future work.

## 5 First order electroweak phase transition plus PQ

In this section we study the effect of the axion string and its portal coupling to the Higgs sector in scenarios where the EWPT is first order. As a proxy for a realistic model we consider the deformation of the SM potential parametrized by the trilinear term $\propto \delta$ in (2.5). We then fix $V_{\text{EW}} = V_\delta$ in this section.

As we are interested in the general picture emerging from the Higgs interaction with the strings, we will fix the (free) parameter $\delta$ as follows: For $T < T_c$ both seeded tunneling around the strings and homogeneous tunneling from the vacuum B in (2.8) to A in (2.7) far from the strings are in principle possible. To simplify our discussion of the thermal history, we will consider the case in which the barrier is large enough that the Universe would actually remain trapped in the EW–preserving false vacuum B at $T = 0$, as a result of a too slow homogeneous nucleation rate. In the parameterization (2.5) this corresponds to $\delta \lesssim -1.5$, so that we will fix $\delta = -1.6$ in what follows.

In this scenario EW symmetry breaking could be successful only thanks to the seeded process starting on the axion strings. We however stress that the main features we are going to discuss will remain the same for other less extreme choices of the barrier $\delta$, where one should in addition compare the seeded and the homogeneous tunneling rates to determine how (and if) the EW phase transition proceeds. Moreover, we argue that our findings will generically apply beyond the parameterization (2.5) to realistic models leading to a first order EW phase transition, such as the Higgs plus Singlet or 2HDMs.

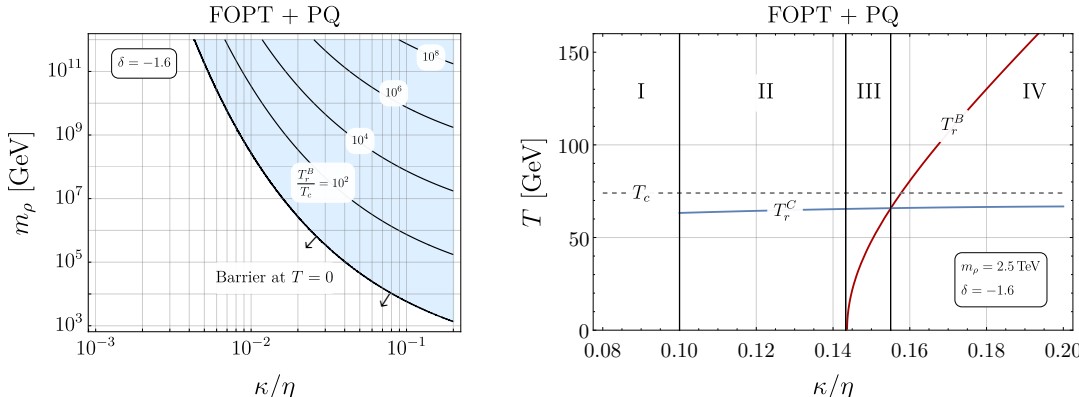

**Figure 6**: **Left:** Contours of the temperature $T_r^B$ at which the string B becomes classically unstable in a model with a first order EW phase transition. In the white region the string B remains (meta)stable down to zero temperature, indicating that the phase transition will necessarily proceed via (seeded) bubble nucleation. **Right:** Identification of the possible cosmological history based on the temperatures at which the different string solutions become unstable (see text for details). In the III and IV zone, the EW symmetry breaking vacuum is eventually reached via classical instability. In the I and II zone, the phase transition should proceed via seeded tunneling.

## 5.1 The paths to electroweak symmetry breaking

The first step to understand how the EW phase transition will proceed in this scenario is to determine what are the stable or metastable string configurations which are encountered during the thermal history of the model. This is because the transitions we are going to consider always involve strings in the initial and final state. In general, all the string solutions A, B, and C can exist and be classically stable for a certain range of temperatures, leading to a rich phenomenology. This is because the deformation $\delta$ in (2.5) allows the scalar potential extrema A and B to be local minima at the same temperature.

Since the standard string B is the one realized at high temperatures, it makes sense to look at the values of $\kappa/\eta$ and $m_\rho$ for which the string B is classically unstable at $T = 0$. This can be easily determined by imposing $\omega^2(T = 0) \leq 0$ in (3.12), and it corresponds to the region of parameter space colored in light blue in the left panel of Fig. 6. In this region, the EW phase transition is guaranteed to complete as there is eventually no barrier preventing the Higgs field to reach the string A configuration corresponding to the true vacuum.

The transition from string B to string A can however occur in different ways depending on $\kappa/\eta$ and $m_\rho$. This can be understood from the right panel of Fig.6, where we identify two different zones, dubbed III and IV, that correspond to the values of $\kappa/\eta$ for which the string B will eventually become unstable at some temperature $T_r^B > 0$ (fixing $m_\rho = 2.5\,\text{TeV}$ for concreteness). Both these zones are contained within the light blue region in the left panel, but they are distinguished due to the behavior of the string C. In fact, this string solution exists only for a certain range of temperatures that we numerically determine to

be $T > T_r^C$ (as indicated by the blue line in the right panel), while for $T < T_r^C$ the string C is no longer supported. By looking at the interplay between $T_r^B$ and $T_r^C$, we can see that in the IV zone the phase transition will proceed in two steps involving different string configurations:

$$\text{IV}: \quad B_{\text{str}} \to C_{\text{str}} \to A_{\text{str}}, \tag{5.1}$$

where we have added the suffix to stress that these transitions have strings as initial and final states. This path follows from the fact that $T_r^B > T_r^C$, and each of the transitions above is expected to be (perhaps weakly) first order as it involves thermal barriers that exist only for a certain range of temperatures. Notice that the first transition in (5.1) is strictly lower dimensional as it induces only a local modification of the string B into string C, and it does not affect the region of space far from the strings. This transition can then occur independently of whether the temperature is above or below the critical temperature $T_c$ (defined in Sec. 2.5 in terms of the EW vacua A and B). The second step can instead only occur below $T_c$. In particular, as soon as $T \lesssim T_r^C < T_c$ the string C becomes classically unstable and, as we shall describe in Sec. 5.2, it dynamically evolves into string A leading to the completion of the EW phase transition everywhere in space.

As for the III zone, there exist two possible paths involving either a two–step or a one–step transition between strings,

$$\text{III}: \quad B_{\text{str}} \rightsquigarrow C_{\text{str}} \to A_{\text{str}}, \quad \text{or} \quad B_{\text{str}} \to A_{\text{str}}, \tag{5.2}$$

In the first path the $B \rightsquigarrow C$ step is again strictly lower dimensional. However in this case it is a tunnelling process for which the barrier does not necessarily vanish, and it is therefore denoted with a wiggled arrow $\rightsquigarrow$ to distinguish it from a process which is instead guaranteed to take place at low enough temperatures (represented by a straight arrow $\to$). The second path is the one relevant for transitions happening at $T < T_r^C$ when the string C solution is no longer supported. The transitions ending on the A string involve the whole 3D space, and are determined by thermal barriers that are vanishing at $T_r^C$ when the string develops a classical instability. Which of the two paths in (5.2) will be realized depends on the $B \rightsquigarrow C$ tunneling rate and needs to be determined case by case.

Let us now consider the other possible scenario in which the string B is metastable at $T = 0$. This corresponds to $\omega^2(T = 0) > 0$ in (3.12). In this case EW symmetry breaking may or may not be successful depending on the actual rate of seeded tunneling. This region of parameter space is shown in white in the left panel of Fig. 6. It includes two possible sub–cases which are labeled by I and II zone in the right panel of Fig. 6. The difference is that in the I zone there is no range of temperatures for which the string C can be found, whereas in the II zone the string C exists for $T > T_r^C$.

For points in the I zone the only possible transition is then:

$$\text{I}: \quad B_{\text{str}} \rightsquigarrow A_{\text{str}}. \tag{5.3}$$

Since here the barrier between string B and A survives at zero temperature, this transition can in principle be significantly supercooled.

In the II zone there are instead two possibilities in analogy to (5.2),

$$\text{II}: \quad B_{\text{str}} \rightsquigarrow C_{\text{str}} \to A_{\text{str}}, \quad \text{or} \quad B_{\text{str}} \rightsquigarrow A_{\text{str}}, \tag{5.4}$$

where the second path is the relevant one for temperatures $T < T_r^C$. Here again we denoted by $\rightsquigarrow$ a tunneling process which may not occur in the expanding universe depending on the rate, while by $\to$ we denote a transition necessarily ending due to classical instability. The crucial difference between (5.4) and (5.2) is that transitions proceeding via the second path in (5.4) may not lead to successful nucleation depending on the rate of seeded tunneling.

To summarize, we can divide the values of $\kappa/\eta$ at fixed $m_\rho$ in two different regimes: i) zone I and II where there is a barrier for completing the (seeded) EW phase transition down to zero temperature, so that these points are viable only if seeded tunneling is fast enough; ii) zone III and IV, where the axion string becomes classically unstable during the cosmological evolution and guarantees the completion of the EW phase transition. Scenarios ii) allow for a potentially large range of temperatures $T < T_r^B$ in the early Universe with broken EW symmetry inside the strings. On the other hand, scenarios i) can allow for some degree of supercooling at the time of the EW phase transition.

Let us finally comment on how this picture can change when considering different values for the potential barrier between the vacua A and B. For larger barriers (more negative $\delta$) homogeneous tunneling is even more suppressed, and string C can potentially be metastable down to zero temperature even in zone II. For smaller barriers (less negative $\delta$), the EW phase transition can in principle complete via homogeneous tunneling as well, and one needs to check which process is actually faster. Clearly, for sufficiently small $\kappa/\eta$ the strings decouple and seeded tunneling is suppressed. On the other hand, for values of $\kappa/\eta$ and $m_\rho$ such that $T_{\text{roll}}^B > T_c \sim T_{\text{roll}}^C$, the phase transition is rapidly completed by the classical instability of the axion string, and homogeneous tunneling is likely to be superseeded. Scenarios in between these two limits require a quantitative analysis.

In the next sections we will focus on the detailed dynamics of EW symmetry breaking, and consider two benchmark points: one belonging to the IV zone (Sec. 5.2) for which the transition is controlled by classical instability, and one in the I zone (Sec. 5.3) where the transition proceeds via seeded tunneling.

## 5.2 Classical instability (rolling)

We here focus on a representative benchmark point belonging to the IV zone of Fig. 6, where the path to electroweak symmetry breaking follows Eq. (5.1). This is specified by the value of $\kappa/\eta = 0.18$ and $m_\rho = 2.5\,\text{TeV}$.

The thermal history is as follows: at a certain temperature $\bar{T} \gtrsim T_r^B \simeq 130\,\text{GeV}$ the string B is transformed into string C via a lower dimensional phase transition which is expected to be weakly first order. Notice that this occurs above the critical temperature for the EW vacua given here by $T_c \simeq 74\,\text{GeV}$.

When the temperature reaches $T = T_c$ the string network consists of C strings, and for $T < T_c$ the decay into A strings becomes energetically viable. On the other hand, the string C is metastable only for $T > T_r^C$, so that there exists only a very small range of temperature

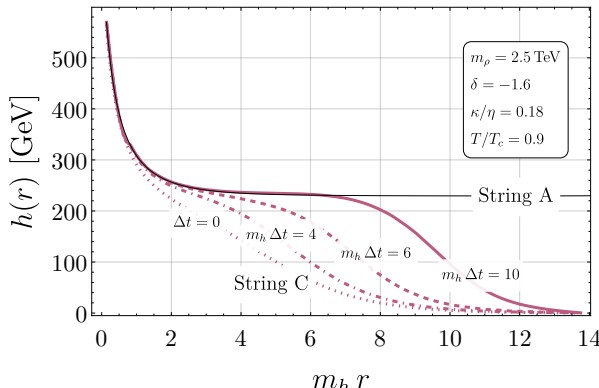

**Figure 7**: An example of a *rolling* process. The unstable string C configuration (dotted red line) is evolving into string A (solid black line). We show snapshots at different times (dash-dotted, dashed, thick red) to illustrate the time evolution of the string profile.

when seeded tunneling can take place. This suggests a very fast transition in which the whole string C starts evolving at the same time towards the string A configuration [6].

This is shown in Fig. 7, where we solve the real time evolution of the Higgs field by using the C profile as initial condition ($\Delta t = 0$) for the motion at temperatures slightly below $T_r^C$. Since at this temperature the C string is unstable, the profile evolves towards the string A configuration shown by the solid black line. The dynamics is solved according to the effective theory (3.8) and we show snapshots at intervals $\Delta t \sim 1/m_h$.

It is important to notice that this time–dependent Higgs profile behaves effectively as an infinitely–long cylindrical bubble wall which expands along the direction orthogonal to the string close to the speed of light (as we are neglecting any possible source of friction in our equation of motion), converting the whole space to the true vacuum similarly to a standard first order transition. A detailed study of the phenomenological consequences and applications of this type of bubble walls is left for future work.

### 5.3  String tunneling into a new string

In this section we study the cosmological history of the I zone in Fig. 6 (right), where the axion strings act as seeds to catalyze bubble nucleation during the EW phase transition. The relevant process is the seeded tunneling from string B to string A depicted in (5.3).

There are several differences with respect to standard homogenous tunneling. First, the relevant rate here is the nucleation rate per unit string length, as opposed to the rate per unit volume. This yields a modified (stricter) nucleation condition in the expanding universe. Secondly, the computation of the tunneling profile is more involved as the system has a reduced symmetry compared to the $O(3)$ homogeneous space, with only $O(2)$ rotational symmetry on the plane orthogonal to the string. We will tackle this computation with different approaches depending on the regime of $\kappa/\eta$, as we shall see in detail.

---

[6]This process has been described also for DW seeds as *rolling* in [33].

### 5.3.1 Nucleation condition

We define the thermal tunneling rate per unit length as

$$\gamma_{\mathrm{s}} \simeq T^2 \exp(-S_{\mathrm{string}}/T) \tag{5.5}$$

where $S_{\mathrm{string}}$ is the euclidean action evaluated on the bubble solution interpolating between string B and string A, and the prefactor is estimated as $T^2$ by simple dimensional analysis.

Indicating by $\xi$ the number of strings per Hubble volume, the nucleation condition for the seeded phase transition in the expanding universe reads

$$\mathcal{N}(T_n) = \int_{T_n}^{T_c} \xi \frac{\gamma_{\mathrm{s}}}{H^2} \frac{dT}{T} \simeq 1, \tag{5.6}$$

which corresponds to

$$\frac{S_{\mathrm{string}}}{T} \simeq 2\log(M_{\mathrm{Pl}}/T_n) + \log\xi - 2.4 \approx 73, \tag{5.7}$$

where in the last step we have considered $\xi \sim O(1)$ and $T_n \sim O(100)$ GeV. In the scenario we discuss, the action $S_{\mathrm{string}}$ will be associated to a non spherical bubble nucleated around the string.

### 5.3.2 Tunneling rate

In order to evaluate the rate of seeded tunneling, one needs to find the saddle point of a system of partial differential equations (PDEs) involving the radial PQ mode $\rho(x^\mu)$, the PQ phase $\alpha(x^\mu)$ and the Higgs field $h(x^\mu)$. Considering the high–temperature limit where the tunneling is independent of the euclidean time, and adopting cylindrical coordinates $(r, \theta, z)$, with $z$ aligned with the string, one has

$$\partial_z^2\rho + \partial_r^2\rho + \frac{1}{r}\partial_r\rho - \frac{\rho}{r^2} = \frac{\partial V(\rho, h)}{\partial\rho}, \quad \partial_z^2 h + \partial_r^2 h + \frac{1}{r}\partial_r h = \frac{\partial V(\rho, h)}{\partial h}. \tag{5.8}$$

The equation for the phase $\alpha$ is automatically satisfied during the tunneling event if one takes $\alpha = \theta$ and $\rho = \rho(r, z)$. The boundary conditions are such that

$$\rho(r \to \infty, z) = \tilde{f}_a, \quad \rho(r, |z| \to \infty) = \rho(r), \quad h(r \to \infty, z) = 0, \quad h(r, |z| \to \infty) = 0, \tag{5.9}$$

so that the system approaches the false vacuum B, which now contains the unperturbed string B indicated by $\rho(r)$, far from the nucleation point, namely for $r, |z| \to \infty$. In addition

$$\rho(r = 0, z) = 0, \quad \partial_z\rho|_{z=0} = 0, \quad \partial_r h|_{r=0} = 0, \quad \partial_z h|_{z=0} = 0. \tag{5.10}$$

One can also derive the tunneling equations directly from the EFT description in (3.8). In this case the problem is simplified to a one–field tunneling process:

$$\partial_z^2 h + \partial_r^2 h + \frac{1}{r}\partial_r h + (\kappa/\eta)\frac{h}{r^2} = V'_{\mathrm{EW}}(h), \tag{5.11}$$

where the boundary conditions are now taking into account the presence of the string inside the bubble at $r \leq \epsilon$:

$$\epsilon \, \partial_r h|_{r=\epsilon} = -\frac{\kappa}{\eta} C(\epsilon) h|_{r=\epsilon}, \quad \partial_z h|_{z=0} = 0, \quad h|_{|z|=\infty} = 0, \quad h|_{r=\infty} = 0. \tag{5.12}$$

Clearly for $\kappa = 0$ the equation of motion (5.11) and the boundary conditions (5.12) become spherically symmetric, and seeded tunneling reduces to homogenous tunneling with $O(3)$ symmetry. On the other hand a non–vanishing portal implies asymmetric motion due to the centrifugal term in (5.11) as well as asymmetric boundary conditions with possibly strong gradients at $r = \epsilon \sim 1/m_\rho$.

An exact numerical evaluation of (5.8) or (5.11) is beyond the scope of our analysis, and we employ several approximations to compute the seeded bounce action, as we discuss below.

**Numerical PDE solution for the Higgs perturbation**   In order to find the bounce profile numerically, we exploit the fact that the bounce solution should reduce to the homogeneous bounce with $O(3)$ symmetry in the decoupling limit of small $\kappa/\eta$. The strategy consists in first finding the homogenous $O(3)$ profiles that would be exact for $\kappa = 0$, for instance with the help of `FindBounce`[88], and then solve (5.8) by perturbing around these homogeneous profiles. In doing so we actually solve only the partial differential equation for the Higgs field in (5.8), and we treat the $\rho$ profile as frozen to the string B. This corresponds to ignore backreaction effects which are expected to be small given the small portals we are interested in, and the fact that $m_h/m_\rho \ll 1$. Equivalently, one could try to directly solve (5.11) with the appropriate boundary conditions.

Parametrizing the Higgs bounce as the $O(3)$ homogeneous solution plus a deformation $\delta h(r, z)$ we find that for small $\kappa/\eta$ the built–in numerical solver in `Mathematica` converges to a solution for $\delta h$, which we then use to compute the corresponding bounce action.

**Analytic bounce action at the linear order in the EFT**   We also provide a semi–analytic derivation of the seeded tunneling action by following an expansion in small $\kappa/\eta$, namely for $\kappa/\eta \ll (\kappa/\eta)_c$, where the latter is defined as the value of the portal for which the string B becomes unstable at a given temperature.

In this limit, the effect of the B string can be considered as a small correction to the homogenous tunneling solution which respects the $O(3)$ symmetry. The idea is again to expand the Higgs profile around the homogeneous solution:

$$h(z, r) = h_{\rm h}(\xi) + \frac{\kappa}{\eta} \delta h(r, z), \quad \xi = \sqrt{r^2 + z^2}. \tag{5.13}$$

The expansion aims to capture the leading order in $\kappa/\eta$, and takes advantage of the fact that the string–independent part of the effective action (3.8) is extremal at $h_{\rm h}$. We then have:

$$S_{\rm string}[h_{\rm h} + \delta h] = S_{\rm hom}[h_{\rm h}] + \delta S[h_{\rm h}, \delta h] \tag{5.14}$$

with

$$\delta S = -\pi \frac{\kappa}{\eta} \int dz \int_\epsilon^\infty r \, dr \left\{ \frac{1}{r^2} + 2\pi C(\epsilon) \delta^{(2)}(r - \epsilon) \right\} h_{\rm h}^2(\xi) + \mathcal{O}(\kappa^2). \tag{5.15}$$

As we can see, the linear order contribution in the portal coupling is actually independent of $\delta h$, and the correction to the homogenous tunneling due to the string can be evaluated straightforwardly from (5.15) once the $O(3)$ symmetric bounce profile is known.

We can further simplify our expression for $\delta S$ assuming a typical shape for the homogenous bounce solution, namely a thin–wall spherical bubble of radius $R$ and release point in the interior $h_{\rm h}(0)$. The integral (5.15) reduces to

$$\delta S_{\rm TW} = -2\pi R \frac{\kappa}{\eta} \left[ \log\left(\frac{2R}{\epsilon}\right) + C(\epsilon) - 1 \right] h_{\rm h}(0)^2 \equiv 2\pi R \, \Delta\mu_{\rm eff}. \tag{5.16}$$

As we can see, the string induces a change in the tunneling action which can be recasted as the energy difference due to a change in tension of the axion string between the initial and final state. From this result one may be tempted to go backwards and actually start from a thin–wall expression for the energy of an approximately spherical bubble of radius $R$ around the string,

$$E(R) = 4\pi R^2 \sigma - \frac{4\pi}{3} R^3 \epsilon - 2\pi R \Delta\mu. \tag{5.17}$$

where $\sigma$ and $\epsilon$ are the usual tension and vacuum energy related to the homogenous bubble. It is however not obvious what the correct choice for $\Delta\mu$ is, as the tension in (5.16) is not the same as for instance the difference between string B and string A given in (3.19) (even though they share a similar structure). In particular, the value of the Higgs at the center of the homogenous bubble $h_{\rm h}(0)$ does not have to coincide with the core of string A [7].

In the following we shall then use (5.16) to estimate the effect of the axion string in the small portal limit, as this is derived directly from the effective action.

**Bounce action in the lower dimensional theory**   Let us now consider the opposite case in which the portal coupling is large enough that the seeded bubble differs very much from a sphere, $\kappa/\eta \lesssim (\kappa/\eta)_c$.

In this limit, we can no longer expand around the homogenous trajectory $h_{\rm h}$. We may however refer to the reduced theory living on the string characterized by the effective action (3.21). Seeded tunneling is then described in terms of the mode $h_0(z, t)$. For this to be successful we need the effective potential in (3.21) to develop a minimum with $\tilde{V}(h_0) \leq 0$ away from the origin. This makes sure that $h_0 \equiv 0$, which corresponds to the unperturbed string B in this description, is not a global minimum of the theory.

Notice that since the effective potential $\tilde{V}$ is obtained by neglecting all the Higgs excitations but the lightest, the results obtained this way are accurate only when some hierarchy exists between $\omega^2$ and the 4d Higgs mass $m_h^2$, namely $\omega^2/m_h^2 \ll 1$.

Considering a tunneling event driven by thermal fluctuations, we may search for time–independent tunneling solutions $h_0(z)$, so that our problem becomes effectively one dimensional[8]. The corresponding action is then simply obtained as

$$S_{\rm string} = 2 \int_0^{h_{0,r}} \sqrt{2\tilde{V}(h_0)} \, dh_0, \quad h_{0,r} : \tilde{V}(h_{0,r}) = 0, \tag{5.18}$$

---

[7]This would be the case only very close to the decoupling limit, $\kappa/\eta \ll 1$, where $h(0) \simeq v$, see Fig. 4.

[8]When considering quantum fluctuations as the main driver, one may still use the same effective potential $\tilde{V}(h_0)$ and search for $O(2) \times O(2)$ solutions $h(\rho, r)$ with $\rho^2 = z^2 + \tau^2$ and $t = i\tau$.

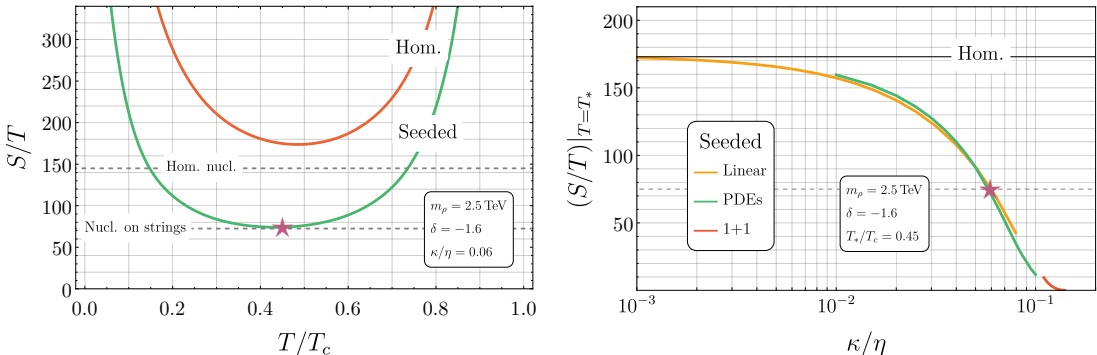

**Figure 8**: **Left:** The homogenous and seeded bounce action as a function of the temperature for the selected benchmark with $\kappa/\eta = 0.06$. In red the homogeneous tunneling bounce, which is too large to lead to successful nucleation. In green the string seeded bounce action, which successfully catalyzes the EW phase transition at $T_n \simeq 0.45\,T_c$, marked with a red star in the plot. **Right:** Bounce actions evaluated at $T \simeq 0.45\,T_c$. The bounce action for the tunneling of string B into string A is shown as a function of $\kappa/\eta$ for $m_\rho = 2.5\,\mathrm{TeV}$ by employing different approximations as described in the text: in orange the analytic approximation valid for small values of $\kappa/\eta$, in green the numeric solution of the Higgs PDE, in red the bounce action obtained in the 1+1 theory on the string. The homogeneous bounce action is shown for comparison by the solid black line.

where the factor of two comes from the symmetry around $z = 0$.

As by assumption $\omega^2/m_h^2 \ll 1$, seeded tunneling is very much enhanced compared to homogeneous tunneling, given that $\omega^2$ sets the size of the potential barrier. In this case the shape of the bubble deviates significantly from spherical symmetry and becomes a prolate ellipsoid. In our analysis we have evaluated (5.18) numerically. We however observe that for really small $\omega^2$ the bounce action can be further simplified by neglecting the quartic coupling in $\tilde{V}$, leading to a semi–analytic expression which goes very rapidly to zero with $\omega$, namely $S_{\mathrm{inh}} \propto \omega^5$.

**Results for string–seeded tunneling** Making use of the computational tools presented above, we can now present some results for the tunneling seeded by the axion string.

We considered a benchmark with a moderate hierarchy between $m_h$ and $m_\rho$ so that numerical routines are stable, but the qualitative conclusions will be generic. In particular we specify $m_\rho = 2.5\,\mathrm{TeV}$ and consider the range $10^{-3} \lesssim \kappa/\eta \lesssim 0.1$, namely the I zone in the right panel of Fig. 6.

In Fig. 8 (left) we display the bounce action as a function of the temperature for $\kappa/\eta = 0.06$. The homogeneous bounce action is shown by the orange line, and it is too suppressed to lead to successful nucleation. On the contrary, on the selected benchmark for $\kappa/\eta$ the string–seeded tunneling rate (computed with the linear approximation and by solving the corresponding Higgs PDE) is large enough to satisfy the nucleation condition in (5.7) and lead to successful nucleation at $T/T_c \simeq 0.45$, marked with a red star in the Figure.

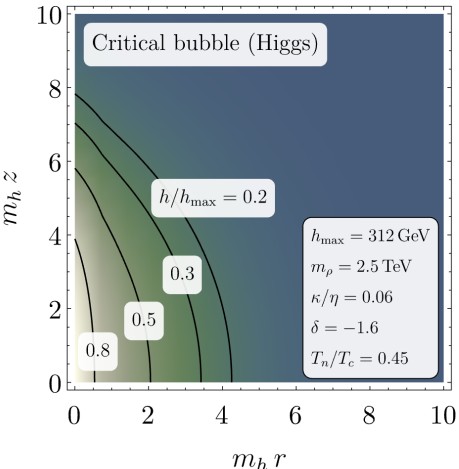

**Figure 9**: Contours of the Higgs field corresponding to the critical bubble for the string–seeded bounce, evaluated at the nucleation temperature $T_n \simeq 35$ GeV for the benchmark $m_\rho = 2.5$ TeV, $\delta = -1.6$ and $\kappa/\eta = 0.06$, obtained by solving the Higgs PDE. The contours highlight the non spherical nature of the bubble, elongated along the direction of the string, which sits at $r = 0$ and extends vertically along $z$. The value of the Higgs field in the interior of the bubble (release point) very close to the string core is actually larger than $v = 246$ GeV indicating that this tunneling event is partially reconstructing the profile of string A, which has a Higgs core of about 350 GeV in this benchmark.

Notice that this temperature approximately corresponds to the maximal homogeneous tunneling rate (which is still too slow for successful nucleation).

In Fig. 8 (right) we show the bounce action for the seeded phase transition computed according to the three different methods illustrated above, in the appropriate regime of validity. We see that the three methods nicely complement each other in providing the complete picture for the seeded bounce action. We selected as a representative temperature the value $T/T_c \simeq 0.45$, which corresponds to nucleation for $\kappa/\eta = 0.06$. In the same plot, we show for comparison the value of the homogenous bounce action at this temperature, which is independent of $\kappa/\eta$.

The shape of $S/T$ as a function of $\kappa/\eta$ shows some of the features that we have already encountered in the previous sections. In particular, for $\kappa/\eta \lesssim 10^{-2}$ the string effectively decouples and it can no longer influence the EW phase transition. As a consequence, the seeded bounce action reduces to the homogenous one. On the other hand, for $10^{-2} \lesssim \kappa/\eta \lesssim 0.1$ seeded nucleation is very fast and catalyzes efficiently the EW phase transition. These values of $\kappa/\eta$ are in fact close to the classical instability (occurring here at $\kappa/\eta \approx 0.15$) and the barrier for seeded tunneling is significantly suppressed.

In order to characterize the features of the seeded phase transition, we can further inspect the shape of the critical bubble focussing on the red–star benchmark of Fig. 8. In Fig. 9 we show the Higgs profile corresponding to the seeded bubble nucleated around the

string at $r = 0$. The Higgs is zero far from the string, while it develops a non vanishing expectation value inside the bubble. As we can see, the bubble has a non spherical shape elongated along the string direction. In addition, we notice that the value of the Higgs at the center of the bubble, namely the release point, is actually larger than $v = 246\,\text{GeV}$. This can be understood by noticing that the Higgs field is partially reconstructing the profile of string A (with a large EW breaking condensate) in the interior of the bubble close to the string core.

The 3D representation of this bubble can be seen in Fig. 1. There, we display in red the string B profile given by $\rho(r)$, while the green contour identifies the surface with $h(r) \sim 25\,\text{GeV}$ to illustrate the bubble size and shape.

## 6   Conclusion

We have studied the impact of QCD axion strings at the time of the electroweak phase transition in the most minimal KSVZ axion solution to the strong CP problem. The analysis is carried out as a function of the (only) portal coupling between the PQ and the Higgs sector. Such coupling is generically present at the tree level, but can also be generated via loops of KSVZ fermions. In our study we have been agnostic about the origin of the portal, which has been treated essentially as a free parameter.

We have found that the effect of the axion string on the electroweak sector is actually controlled by the ratio between the portal coupling and the PQ self interaction, $\kappa/\eta$, as well as the mass of the PQ radial mode, $m_\rho$. Our results show that, due to a strong exponential dependence on the portal, axion strings will decouple for all realistic values of $m_\rho$ as long as $\kappa/\eta \lesssim 10^{-3}$. This decoupling value is due to the global nature of the axion string, while local strings would actually decouple much faster.

On the other hand, for larger values of $\kappa/\eta$ axion strings can strongly affect the thermal history of electroweak symmetry breaking. This can happen in two distinct ways. First, even in the pure SM plus PQ theory there can be a large range of temperatures above the QCD scale where the axion strings develop a Higgs condensate at their core; this can for instance affect the numerical simulations aiming at understanding the detailed dynamics of QCD axion strings in the early universe.

Secondly, in BSM models where the EW phase transition is first order, the presence of the axion strings can modify the way the EW phase transition proceeds, either by developing a classical instability (rolling) or by catalyzing bubble nucleation. Such a string–driven electroweak phase transition comes with new phenomenological features, such as cylindrical bubbles of true vacuum expanding radially from the string core, or the nucleation of elongated bubbles nucleated along the string. This can drastically change the expected gravitational wave signal (for instance due to the shape of the bubbles) as well as possible predictions for baryogenesis (due to different regimes for the wall velocity).

Our results have been conveniently obtained within an effective–field–theory approach taking advantage of the hierarchy between the electroweak and the PQ scale, in which the axion string is integrated out at tree level together with the heavy states of the PQ sector. This allows us to obtain analytical results for the stability of the axion string,

as well as to provide a simpler picture of seeded nucleation around heavy defects. This framework can be straightforwardly generalized to a richer electroweak scalar sector beyond the simple deformation of the SM potential considered here, thus paving the way to new phenomenological applications and interesting revisitations of (extensions of) the SM when considered in combination with the axion solution to the strong CP problem.

Let us finally mention that while we have restricted our study to KSVZ–like models where the Higgs is neutral under the PQ symmetry, we expect similar implications for the electroweak phase transition also in DFSZ–like models where the Higgs doublets have additional couplings with the string due to the non–zero PQ charge.

## Acknowledgments

We thank Yu Hamada, David Mateos, Alex Pomarol, Oriol Pujolas, Fabrizio Rompineve, and Miguel Vanvlasselaer for useful discussions. SB is supported by the Deutsche Forschungsgemeinschaft under Germany's Excellence Strategy - EXC 2121 Quantum Universe - 390833306. SB is supported in part by FWO-Vlaanderen through grant numbers 12B2323N. SB and AM are supported in part by the Strategic Research Program High-Energy Physics of the Research Council of the Vrije Universiteit Brussel and by the iBOF "Unlocking the Dark Universe with Gravitational Wave Observations: from Quantum Optics to Quantum Gravity" of the Vlaamse Interuniversitaire Raad. SB and AM are also supported in part by the "Excellence of Science - EOS" - be.h project n.30820817, and by the FWO-NSFC samenwerkingsproject VS02223N.

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
