# Peer review of "QCD Axion Strings or Seeds?"

_SciPost Physics, doi:SciPost Phys. 18, 016 (2025)_

## Round 1 · Referee Report · Anonymous (Referee 1) · 2024-8-7

Report

The Authors of this article studied for the first time how the QCD axion strings impact the evolution of the electroweak phase transition. They focus on both the case in which the EW sector is the Standard Model one, as well as the one of theories Beyond the Standard Model with a first order phase transition.

In the minimal KSVZ model, where a quartic interaction between the Peccei Quinn complex scalar and the Standard Model Higgs doublet is possible, there is an effective coupling between the Higgs field and the axion strings. The effect of the strings on the electroweak phase transition is studied as a function of this effective interaction. The Authors determined the conditions under which the axion string can develop a sizeable Higgs core, where the EW symmetry is broken at temperatures above the EW scale. They found that the effect is controlled by the ratio between the portal coupling and the Peccei Quinn self interaction and by the mass of the radial mode. This can modify the dynamics of the QCD axion strings in the early universe.

Moreover, the Authors consider beyond the Standard Model theories with first order electroweak phase transitions and show that the axion strings may modify the electroweak phase transition in two ways: by catalysing bubble nucleation or by developing a classical instability. This has an impact on predictions for baryogengesis and gravitational wave detection.

This submission meets all of the general acceptance criteria and shows an important result on the effect of QCD axion strings on the cosmological history of the electroweak symmetry breaking and it deserves publication.

Requested changes

Comments the Authors may decide to address:
1 - Can the Authors give an estimate of the effect on the numerical simulation for the detailed dynamics of QCD axion strings in the early universe?
2 - How realistic is the parameterisation for the EFT in real models?
3 - How would the effects of QCD axion string change in DFSZ models?

Recommendation

Publish (meets expectations and criteria for this Journal)

  • validity: -
  • significance: -
  • originality: -
  • clarity: -
  • formatting: -
  • grammar: -

Author:  Simone Blasi  on 2024-10-21  [id 4881]

(in reply to Report 1 on 2024-08-07)
Category:
answer to question

1 - Due to the presence of the large Higgs condensate in the type C string solution, we expect a new contribution to particle friction from the SM bath on the string network that is active from the temperature where the condensate forms until a “friction temperature” that can be estimated to be around the electroweak scale depending on the axion decay constant. This friction can induce large deviations from the scaling regime that can affect the final prediction for the axion relic abundance. Moreover, the axion-Higgs portal combined with the non-trivial Higgs profile inside the string can provide an additional production channel for axion particles. We have added a text explaining these reasonings at the end of Section 4.

2 - We expect this parameterization to work in the case of large hierarchies between the Higgs and PQ radial mode mass, together with kappa/eta < 1, as stated at the beginning of Section 3. We have checked that the EFT reproduces very well quantitative results in the complete theory, in particular for all the string solutions. In addition, we have also found agreement between the linear approximation to the seeded bounce action evaluated within the EFT and the full PDE solution to the tunneling in the complete theory. Nevertheless, it would be interesting to systematically study the effects of higher-order corrections to the EFT presented in the paper.

3 - We expect some changes due to the fact that the Higgs is now charged under the PQ symmetry, presumably captured by additional operators in the EFT. We have added a comment on this at the end of section 6.

---

## Round 1 · Referee Report · Anonymous (Referee 2) · 2024-9-9

Report

The study investigated the impact of QCD axion strings during the electroweak phase transition in the most minimal KSVZ axion solution to the strong CP problem. The publication criteria are amply met, and I recommend the publication of the study.

Requested changes

1- Throughout the work, it is repeatedly emphasized that the value of the portal coupling is treated as a free parameter, and a deliberately agnostic stance is taken regarding its origin. However, in the absence of QCD interactions, it seems reasonable to state that a small value of the portal coupling is technically natural, as in the limit where it approaches zero, an enhanced Poincaré symmetry would be restored (e.g. 1310.0223). Consequently, since the effects of QCD are 3-loop suppressed, one could conclude that it is natural to expect the portal coupling to be small. I would like to ask the authors whether this reasoning is correct. In that case, perhaps the discussion in the 'Setup' section could be expanded slightly.

2-Regarding the case of the pure SM + PQ theory, is it possible to determine whether the presence of the PQ scalar field contributes to the stability of the electroweak vacuum (e.g., 1203.0237)? It could provide further motivation to consider this case.

3-The phenomenological consequences of the SM + PQ case are somewhat briefly addressed at the end of Section 4. Could the authors provide a more quantitative justification for why one should expect that the spontaneous breakdown of the electroweak symmetry could potentially modify the QCD axion string dynamics in the early universe?

Recommendation

Publish (easily meets expectations and criteria for this Journal; among top 50%)

  • validity: good
  • significance: good
  • originality: good
  • clarity: high
  • formatting: excellent
  • grammar: excellent

Author:  Simone Blasi  on 2024-10-21  [id 4882]

(in reply to Report 2 on 2024-09-09)

1- We agree that the portal is technically natural if QCD (as well as possible mixing between the KSVZ fermions and the SM quarks) is turned off. We however emphasize that the size of kappa is not what determines the physical effect, but rather the ratio between the portal and the PQ self quartic, which can be both naturally small. We have updated our discussion at page 4 accordingly including footnote 4.

2 - The portal interaction can in fact stabilize the SM vacuum thanks to threshold/running effects, which can indeed be taken as motivation to consider a sizable portal. We have included a comment on this at the beginning of section 6.

3 - Due to the presence of the large Higgs condensate in the type C string solution, we expect a new contribution to particle friction from the SM bath on the string network that is active from the temperature where the condensate forms until a “friction temperature” that can be estimated to be around the electroweak scale depending on the axion decay constant. This friction can induce large deviations from the scaling regime that can affect the final prediction for the axion relic abundance. Moreover, the axion-Higgs portal combined with the non-trivial Higgs profile inside the string can provide an additional production channel for axion particles. We have added a text explaining these reasonings at the end of Section 4.

---

## Round 2 · List of Changes

- Added footnote 4 on the naturalness of the PQ-Higgs portal
- Expanded discussion on the phenomenological implications of the Higgs condensate inside the string (end of section 4)
- Comment added on the possible modifications for the DFSZ model (section 6)
- Comment added on the connection between the PQ-Higgs portal and the metastability of the SM vacuum (section 6)

---

## Editorial Decision

published